# Epidermal Loss of RORα Enhances Skin Inflammation in a MC903-Induced Mouse Model of Atopic Dermatitis

**DOI:** 10.3390/ijms241210241

**Published:** 2023-06-16

**Authors:** Xiangmei Hua, Conrad Dean Blosch, Hannah Dorsey, Maria K. Ficaro, Nicole L. Wallace, Richard P. Hsung, Jun Dai

**Affiliations:** 1School of Pharmacy, University of Wisconsin, Madison, WI 53705, USA; xhua23@wisc.edu (X.H.); hannah.dorsey1999@gmail.com (H.D.); mficaro@wisc.edu (M.K.F.); nicolewallace2023@u.northwestern.edu (N.L.W.); richard.hsung@wisc.edu (R.P.H.); 2Biomedical Research Model Services, University of Wisconsin, Madison, WI 53705, USA; cblosch@wisc.edu; 3UW Carbone Cancer Center, University of Wisconsin, Madison, WI 53705, USA

**Keywords:** RORalpha, keratinocytes, barrier, atopic dermatitis, skin inflammation

## Abstract

Atopic dermatitis (AD) is a chronic inflammatory skin disease featuring skin barrier dysfunction and immune dysregulation. Previously, we reported that the retinoid-related orphan nuclear receptor RORα was highly expressed in the epidermis of normal skin. We also found that it positively regulated the expression of differentiation markers and skin barrier-related genes in human keratinocytes. In contrast, epidermal RORα expression was downregulated in the skin lesions of several inflammatory skin diseases, including AD. In this study, we generated mouse strains with epidermis-specific *Rora* ablation to understand the roles of epidermal RORα in regulating AD pathogenesis. Although *Rora* deficiency did not cause overt macroscopic skin abnormalities at the steady state, it greatly amplified MC903-elicited AD-like symptoms by intensifying skin scaliness, increasing epidermal hyperproliferation and barrier impairment, and elevating dermal immune infiltrates, proinflammatory cytokines, and chemokines. Despite the normal appearance at the steady state, *Rora*-deficient skin showed microscopic abnormalities, including mild epidermal hyperplasia, increased TEWL, and elevated mRNA expression of *Krt16*, *Sprr2a*, and *Tslp* genes, indicating subclinical impairment of epidermal barrier functions. Our results substantiate the importance of epidermal RORα in partially suppressing AD development by maintaining normal keratinocyte differentiation and skin barrier function.

## 1. Introduction

Atopic dermatitis (AD) is a chronic, pruritic, and inflammatory skin disease [1]. AD has been attributed to two major factors: (a) T helper 2 (Th2)-mediated immune responses featuring high serum IgE levels and hyperactivation of mast cells and eosinophils, and (b) epidermal barrier dysfunctions [2]. Recent genetic studies with human subjects and mice have strongly connected AD etiology with the loss-of-function (null) mutations of the *FLG* gene, which encodes the cross-linking protein filaggrin, critical for cornified envelope (CE) assembly and epidermal barrier functions [3,4]. Consequently, barrier dysfunction has become widely recognized as a leading cause of AD development [5,6].

Normal barrier formation depends on a progressive keratinocyte differentiation process, accompanied by stage-dependent expression of signature structural proteins, such as keratin 5/14 in the basal layer, keratin 1/10 and involucrin in the spinous layer, and loricrin and profilaggrin in the granular layer [7]. This stepwise gene expression relies on specific transcription factors and signaling pathways. In addition to forming a physical barrier, keratinocytes have important innate immune functions by secreting diverse antimicrobial peptides, cytokines, and chemokines in response to various external stimuli. Keratinocyte-derived Th2-promoting cytokines, including thymic stromal lymphopoietin (TSLP), IL-25, and IL-33, are frequently elevated in AD lesions due to a disrupted barrier [8]. TSLP, an IL-7-like cytokine, allows dendritic cells to prime naïve T-cells toward Th2 polarization and promotes pruritus by activating cutaneous sensory neurons [9,10,11]. Therefore, keratinocyte-deregulated immune activities contribute significantly to AD pathogenesis [5,12,13].

The retinoid-related orphan receptors (RORα, RORb, and RORg) belong to the ligand-gated nuclear hormone receptor superfamily and function as transcription factors by binding to ROR response elements, ROREs, within the regulatory regions of their target genes [14,15,16]. RORα is expressed in a temporal and spatial-dependent manner during embryonic development and is critical for Purkinje cell maturation [17,18], bone formation [19], lymphocyte development [20,21], lipid metabolism [22], and inflammation [23,24,25]. In addition, RORα is a well-established component of the circadian clock, with Bmal1 as one of its best-studied targets [26,27]. Recent studies have identified RORα as a regulator of the development of multiple types of immune cells, including type 2 innate lymphoid cells (ILC2) [28] and T help 17 (Th17) cells [29,30]. Malfunctions in these cells are associated with the pathogenesis of allergic inflammation and autoimmune diseases [30,31].

RORα is highly expressed in human and mouse skin [32,33,34,35], specifically in the suprabasal layers of the epidermis, sebaceous glands, and hair follicles [35]. Our earlier studies have demonstrated that among the four RORα isoforms, RORα4 is prominently expressed by human keratinocytes in a manner that increases with the progression of differentiation [32]. While increasing the levels of RORα4 in HKCs enhanced the expression of structural proteins associated with early and late differentiation and genes involved in lipid barrier formation, *RORA* gene silencing suppressed the expression of these genes. Furthermore, the pro-differentiation function of RORα is mediated at least in part by FOXN1, a well-known pro-differentiation transcription factor that we have established as a novel direct target of RORα in keratinocytes [32]. A recent study using human keratinocytes also revealed RORα’s direct regulation of the expression of CXCL14, an epithelial chemokine involved in the clearance of the skin pathogen *Staphylococcus aureus*. In summary, there is compelling evidence that RORα may be critical for both the differentiation and innate immune activities of keratinocytes [36].

Despite these findings, the in vivo functions of epidermal RORα in skin homeostasis remain largely unknown. Recent reports have shown that epidermal RORα levels are significantly down-regulated in clinical samples of several inflammatory skin diseases, including allergic contact dermatitis, lichen simplex chronicus, sarcoidosis, and AD [37,38]. Our current study addresses the impact of reduced epidermal RORα in cutaneous inflammation. For this purpose, we generated mice with an epidermis-specific knockout of *Rora* to determine their reactions to the topical application of MC903, a low-calcemic vitamin D analog for stimulating keratinocyte TSLP production. We found that epidermal *Rora* deletion was insufficient to trigger spontaneous dermatitis, but it significantly intensified the AD-like symptoms elicited by the MC903 application. In this report, we present our findings.

## 2. Results

### 2.1. Epidermal Rora Expression in Mouse Skin in Response to MC903

To determine the essential roles of RORα in the epidermis, we obtained *Rora^tm1a(EUCOMM)Wtsi^* (*Rora^tm1a/+^*) C57BL/6 embryos with a “knockout first” allele [39,40]. This allele contains a splice acceptor–LacZ cassette inserted upstream of the floxed *Rora* “exon 3” (Figure 1A). We first generated the heterozygous *Rora^tm1b/+^* strain, a LacZ reporter strain with heterozygous *Rora* gene deletion (Figure 1A). Consistent with a previous report [35], *Rora* mRNA was predominantly expressed in the suprabasal layers of the epidermis and sebaceous glands in unchallenged mouse ear skin, as determined by the intensity of the X-gal signal (Figure 1B). At the same time, only a limited signal was detected in the dermis (Figure 1B). In response to the MC903 treatment, the X-gal intensity was substantially reduced in the hyperproliferative epidermis on day 5 and partially restored in the upper layer after day 8. However, no significant change was observed in the dermis (Figure 1B). These results revealed the importance of epidermal RORα for its skin functions under physiological and AD-like pathophysiological conditions.

### 2.2. Epidermal Rora Ablation Exaggerates MC903-Induced Ear Thickening

To verify the essential roles of epidermal RORa in MC903-induced skin inflammation, we generated a *Rora^flox/flox^* strain and a *Rora^LacZΔ/+;^* K14-CRE strain from the *Rora^tm1a/+^* mice (Figure 1A). Crossing of the two strains resulted in mice with wildtype (WT, *Rora^+/fl^*), heterozygous *Rora* null (*Rora^LacZΔ/fl^*), or an epidermal-specific *Rora* knockout (*Rora^LacZΔ/fl^;K14-CRE*), referred to as *Rora^LacZΔ/EKO^*. The epidermal *Rora* gene deletion efficiency was analyzed via genotyping using short-range end-point PCR assays (Figure 1C; Appendix A) and through semi-quantitative RT-PCR analysis of *Rora* mRNA in adult ear skin (Figure 1D). Both young and adult mice with epidermal *Rora* deletion displayed normal hair development and skin appearance at the macroscopic level. However, starting from day 4, the MC903-induced ear thickening was profoundly enhanced in *Rora^LacZΔ/EKO^* mice compared to their WT and heterozygous *Rora^LacZΔ/fl^* littermates (Figure 1E). On day 11, the epidermal *Rora*-deficient mice also displayed severe ear swelling and skin scaliness (Figure 1F), along with greater epidermal hyperplasia and heavy immune infiltration in the dermis, as shown in the hematoxylin and eosin (H&E) staining (Figure 1G). The heterozygous *Rora^LacZΔ/fl^* mice also showed more severe ear swelling than the WT controls starting from day 9 (Figure 1E). The histological analysis also revealed the epidermal and dermal expansions in *Rora* heterozygous mice at a level higher than the WT but lower than the epidermal *Rora* deficient *Rora^LacZΔ/EKO^* mice in the MC903-treated group (Figure 1G). These results substantiate the important roles of epidermal RORα in suppressing MC903-induced skin inflammation in mice.

### 2.3. Epidermal Rora Deletion Aggravates MC903-Induced AD-like Symptoms

Previous reports have identified RORα expression in dermal immune cells, including regulatory T cells (Treg) and group II lymphoid cells (ILC2s) [25,30]. To rule out the contribution of reduced RORα expression from other cell types to the enhanced inflammation shown in *Rora^LacZΔ/EKO^* mice carrying a global heterozygous *Rora* deletion, we established the *Rora^fl/fl^; K14-CRE* strain (referred to as *Rora^EKO^*) to confirm the previous findings. The efficiency of epidermal *Rora* gene deletion was validated through genotyping (Figure 2A; Appendix A) and semi-quantitative RT-PCR analysis of *Rora* mRNA in the back skin epidermis of postnatal day 4 (P4) pups, using the liver tissue as a negative control (Figure 2B). Compared to *Rora^fl/fl^* littermates, MC903-treated *Rora^EKO^* mice exhibited enhanced ear swelling (Figure 2C) that led to severe skin scaliness (Figure 2D), greater epidermal hyperplasia, and heavier dermal infiltrates, including eosinophils and mast cells, as shown in the histological analysis (Figure 2E,F).

Immunostaining further revealed the prominent dermal recruitment of other immune cells in MC903-challenged *Rora^EKO^* skin, including CD11c+ positive dendritic cells, CD4+ T cells, and F4/80+ macrophages, contrasting with the moderate infiltration of these cell types in the *Rora^fl/fl^* control skin (Figure 3A). The hyperproliferative epidermis in *Rora^EKO^* skin also showed higher induction of keratin 6, the marker of inflamed or hyperproliferative keratinocytes, and lower loricrin expression than the control (Figure 3A). Consistent with the heavier dermal infiltration, MC903-stimulated expression of the proinflammatory cytokine *Il1b* and chemokines *Cxcl2* and *Cxcl10* [41] was markedly enhanced by epidermal loss of RORα (Figure 3B). In contrast to the robust local skin inflammation in *Rora^EKO^* mice, MC903-induced systemic Th2 responses, measured by the serum IgE elevation, were slightly increased compared to the control (Figure 3C). These results support the primary contribution of epidermal RORα in suppressing MC903-initiated cutaneous inflammation in mice.

### 2.4. Epidermal Loss of RORα Accelerates MC903-Induced Dermal Activation of Dendritic Cells and CD4+ T Cells

To understand the origin of robust skin inflammation in *Rora^EKO^* mice, we further analyzed AD features at an earlier time point post-MC903 application before the robust takeoff of ear swelling. The level of CD11c+ dendritic cells in *Rora^fl/fl^* mice reached 2-fold and 3.6-fold of the original level at day 2 and day 8 post-MC903 application, respectively. In comparison, *Rora^EKO^* mice exhibited higher elevations, with a 4.5-fold and 7.5-fold elevations of CD11c+ cells (Figure 4A). In contrast to the rapid activation of dermal dendritic cells, MC903-initiated dermal recruitment of CD4+ T cells did not occur until day 8 in *Rora^fl/fl^* mice. This infiltration occurred at least three days in advance, with a distinct enhancement in *Rora^EKO^* mice (Figure 4B). Therefore, epidermal loss of RORα accelerated the MC903-initiated dermal activation of dendritic cells and CD4+ T cells.

### 2.5. Epidermal Rora Deletion Increases Tslp Gene Transcription

MC903-initiated AD-like inflammation relies on keratinocyte production of TSLP [42,43]. An immediate effect of TSLP is to activate skin residential dendritic cells, which could migrate into the ear-draining lymph nodes and prime naïve T cells into Th2 effector cells [11,44,45]. Thus, we studied whether the accelerated activation of CD11c+ dendritic cells in MC903-challenged *Rora^EKO^* skin was due to a higher TSLP production. RT-PCR analysis revealed higher levels of *Tslp* mRNA compared to the *Rora^fl/fl^* controls in both the vehicle- and MC903-treated groups on day 2 (Figure 5A). On the other hand, epidermal *Rora* deletion did not affect the mRNA level of *Cyp24a1*, a direct target gene of VDR’s transcription activity stimulated by MC903 [42,46] (Figure 5A). These results suggest that RORα could regulate *Tslp* gene transcription in keratinocytes via a VDR-independent mechanism.

Despite the higher *Tslp* mRNA expression in *Rora^EKO^* mice (Figure 5A), MC903-induced epidermal and serum production of TSLP protein from day 2 to day 5 were comparable between these mice and the *Rora^fl/fl^* controls (Figure 5B–D). Furthermore, the levels of MC903-stimulated IL13 secretion from ear-draining lymph nodes were comparable between the two genotypes (Figure 5E), which was expected because they should depend on TSLP production from the ear skin in this model [42]. These findings suggest that the enhanced activation of dermal dendritic cells in *Rora^EKO^* mice at the early stages post-MC903 treatment (Figure 4A) was likely due to a TSLP-independent mechanism.

### 2.6. Epidermal Loss of RORα Deregulates Keratinocyte Differentiation and Accelerates Skin Barrier Disruption in Response to MC903

In addition to increased *Tslp* mRNA expression, vehicle-treated *Rora^EKO^* mice also expressed higher mRNA levels of *Krt16*, which encodes keratin 16: the interacting partner of keratin 6 (Figure 6A). Another highly increased gene in unchallenged *Rora^EKO^* skin was *Sprr2a*, which encodes the proline-rich protein 2 (SPRR2), a cross-linking protein in the cornified envelope [47] (Figure 6A). In contrast to its strong effects on *Krt16* and *Sprr2a* expression, epidermal *Rora* deletion had a marginal enhancing effect on *IL1b* and *Cxcl10* expression in both the vehicle-treated and MC903-treated groups (Figure 6A), indicating that these proinflammatory immune factors were more involved in the enhanced inflammatory responses in the *Rora^EKO^* mice at later stages (Figure 3B).

In addition to the elevated expression of *Krt16*, *Sprr2a*, and *Tslp*, vehicle-treated *Rora^EKO^* ear skin showed mild epidermal hyperplasia, which was rapidly enhanced between day 2 and day 5 after MC903 application, along with a robust keratin 6 induction (Figure 6B,C). Although keratin 10 levels were comparable between the two genotypes over this period, involucrin and loricrin levels were markedly lower in the *Rora^EKO^* epidermis than in the *Rora^fl/fl^* control on day 2 (Figure 6B–D; Appendix A). On day 5, while loricrin largely disappeared from the *Rora^EKO^* epidermis, its signal remained intact across the control skin (Figure 6B–D; Appendix A). We also noticed that the involucrin levels in *Rora^EKO^* epidermis was significantly lower than the control levels in the vehicle treatment group (Figure 6C,D; Appendix A). These results substantiate the importance of epidermal RORa in maintaining the expression of late differentiation markers at the steady state and during inflammation. Consistent with the deregulated expression of differentiation markers, *Rora^EKO^* mice also exhibited higher transepidermal water loss (TEWL) rates than the controls before and after the MC903 application, suggesting an accelerated epidermal barrier disruption (Figure 6E). Overall, the overt inflammation in MC903-challenged *Rora^EKO^* mice can be partially attributed to keratinocyte deregulation at the steady state, followed by acceleration in epidermal barrier impairment.

## 3. Discussion

Despite the prevalent RORα expression in the epidermal compartment of normal skin [32,33], the pathophysiological functions of epidermal RORα in AD development have remained underexplored. Adult mice with epidermal *Rora* ablation displayed overall normal skin phenotypes but exhibited severe AD-like symptoms after exposure to MC903. Our data uncovered epidermal RORα’s crucial role in suppressing AD-like cutaneous inflammation.

Genetic studies with human subjects have revealed a strong link between AD etiology and loss-of-function (null) mutations of the *Flg* gene [3]. It has been shown that the *Flg-/-* null mutation alone was insufficient to trigger spontaneous inflammation in mice. Still, this mutation can lower the threshold for cutaneous sensitization to haptens in mouse models with allergic contact dermatitis [48,49]. These investigations confirm barrier dysfunction as a primary cause of allergic cutaneous inflammation. In our study, despite its normal appearance at the steady state, epidermal *Rora*-deficient skin exhibited mild epidermal hyperplasia, greater TEWL rate, and higher mRNA levels of *Krt16*, *Sprr2a*, and *Tslp* compared to the skin of wildtype littermates (Figure 6). Expressions of keratin 16/17 and its partner, keratin 6, are similarly elevated in the hyperproliferative keratinocytes under pathological conditions such as injury and AD [50,51,52]. It is also noteworthy that SPRR2 proteins have been found to be upregulated in mice with impaired barrier functions [53,54,55,56]. More importantly, these barrier dysfunction-related phenotypes in *Rora* deficient skin partially resembled the subclinical conditions found in nonlesional skin of AD patients [52,57]. Therefore, we anticipate that the amplified cutaneous inflammation in MC903-challenged *Rora^EKO^* mice can be partly attributed to the pre-existing epidermal barrier dysfunctions.

Despite the comparable TSLP production between *Rora^EKO^* and *Rora^fl/fl^* mice upon MC903 stimulation (Figure 5), *Rora^EKO^* skin displayed a faster and stronger activation of CD11c+ dendritic cells (Figure 4A). In fact, CD11c+ cells were already detected in *Rora*-deficient skin treated with the vehicle (Figure 4A). A similarly elevated CD11c+ population has also been observed in human skin epidermis carrying *FLG*-null mutations [58]. The increased baseline activation of antigen presentation cells can be attributed to barrier dysfunction and increased exposure to dangerous signals [58]. We postulate that the pre-existing CD11c+ cells or altered microenvironment in the *Rora^EKO^* skin may amplify the TSLP activation of dermal dendritic cells, which can enhance T cell activation at the later stage (Figure 4B).

A recent study demonstrated that epidermal RORα protein expression is significantly lower in human skin lesions of several inflammatory skin diseases than in normal skin [37]. Consistent with this report, using the *Rora* lacZ-reporter mice, we found that *Rora* gene transcription in the granular layers of the epidermis was markedly reduced on day 5 after the MC903 application (Figure 1B), simultaneous with the TSLP peak production (Figure 5B–D). These results implicate epidermal *Rora* as a negative target of the TSLP-initiated inflammatory pathway in the skin. Downregulation of epidermal RORα can further contribute to AD development by accelerating skin barrier impairment, thus perpetuating a vicious cycle. It has been found that elevated Th2 cytokines (IL4 and IL13), a hallmark of AD, broadly suppress the expression of barrier-related genes, such as filaggrin, involucrin, and loricrin, through IL-4Rα-STAT6 signaling [59,60,61]. Along with *Staphylococcus aureus lipoteichoic acid*, Th2 cytokines also repress genes related to early keratinocyte differentiation by modulating the activities of p63 and Notch, the key transcription factors for skin development and keratinocyte differentiation [62]. We believe that epidermal RORα expression is also under the negative control of certain proinflammatory cytokines and downstream signaling pathways.

We have previously reported the effect of SR1001, an inverse agonist of RORα/γ, in suppressing MC903-induced AD-like inflammation [63]. Surprisingly, epidermal loss of RORα had the opposite effect in the same mouse model. This contrast can be attributed to the opposite effects of SR1001 and epidermal *Rora* gene deletion on TSLP gene expression. While SR1001 blocked MC903-induced *Tslp* expression at both the mRNA and protein levels [63], epidermal *Rora* deletion enhanced *Tslp* transcription (Figure 5A). The latter evidence indicates that RORα may function as a transcription repressor of *Tslp* in keratinocytes. As a RORα/γ inverse agonist, binding of SR1001 to RORα can increase the recruitment of co-repressors [64,65]. While this recruitment can downregulate the positive target genes of RORα, it may also enhance the repression of the negative targets, such as *Tslp*. An examination of the anti-inflammatory effects of SR1001 in *Rora^EKO^* mice would further validate the dependency of these effects on epidermal RORα.

A recent report showed that epidermal *Rora* deletion alleviated imiquimod-induced psoriatic-like skin inflammations by attenuating the activation of nuclear factor-κB (NF-κB and STAT3 in keratinocytes) [66]. The opposite effects of epidermal RORα loss on cutaneous inflammation in the two disease models suggest it has context-dependent regulatory functions. While MC903-initiated AD-like responses rely on epithelial production of TSLP, imiquimod-caused psoriatic-like inflammations depend on the activations of Toll-like receptor 7 and subsequent pathways in numerous cell types in mice [67]. Furthermore, while epidermal RORα expression is downregulated in skin lesions of multiple inflammatory skin disorders, including AD, its expression is unaltered in psoriatic skin lesions [37]. We believe that RORα prevents AD development by maintaining normal barrier functions while potentiating psoriatic symptoms through positive regulation of STAT3 activity in imiquimod-treated keratinocytes [66].

Overall, we identified the protective effects of epidermal RORα against MC903-induced AD development. Our findings also revealed the beneficial potentials of RORα agonists to restore barrier functions and alleviate inflammation in treating AD and other inflammatory skin diseases [68]. A strength of this study is that tissue-specific *Rora* knockout mice were used to uncover epidermal RORα’s functions in regulating AD development. Our work differs from most previous studies using the staggerer *Rora^sg/sg^* mice with a global *Rora*-null mutation. Thus far, the underlying molecular mechanisms of epidermal RORα’s pathophysiological functions have not been elucidated. Therefore, identifying keratinocyte-specific targets and upstream regulators of RORα will lead to mechanistic insight into cutaneous inflammation and the discovery of novel therapeutic targets.

## 4. Materials and Methods

### 4.1. Mice

All animal studies were conducted under approved protocols from Institutional Animal Care and Use Committee. *Rora^tm1a(EUCOMM)Wtsi^* (*Rora^tm1a/+^*) embryonic stem cells were obtained from INFRAFRONTIER/EMMA (European Mouse Mutant Archive, Munich, Germany) and subjected to in vitro injection into the C57BL/6 female mice to generate the *Rora^tm1a/+^* mice (performed by Animal Models of Biotechnology at the University of Wisconsin, Madison, WI, USA). A male *Rora^tm1a/+^* mouse was crossed with a female *Sox2* (SRY-box containing gene 2)-*Cre* transgenic mouse (B6.Cg-*Edil3^Tg(Sox2-cre)1Amc^*/J; The Jackson Laboratory, Bar Harbor, ME, USA) [69] to generate the *Rora^tm1b/+^* strain (*Rora^LacZΔ/+^*), which contains a LacZ-cassette upstream the deleted *Rora* exon 3 (Figure 1A). To generate the *Rora^tm1c/+^* (*Rora^flox/+^*) mice, the *Rora^tm1a/+^* mice were crossed with ACTB-FLPe transgenic mice (B6.Cg-Tg(ACTFLPe)9205Dym/J; The Jackson Laboratory) to delete the FRT-flanked section. The *Rora^flox/+^*; *FLP* mice were backcrossed with wildtype C57BL/6 mice to remove FLP [39]. The *Rora^LacZΔ/EKO^* mice were generated by crossing the *Rora^flox/flox^* strain with the *Rora^LacZΔ/+;^K14-Cre* strain (B6N.Cg-Tg(KRT14-cre)1Amc/J, The Jackson Laboratory). The *Rora^EKO^* mice were generated by crossing the *Rora^flox/flox^* strain with the *K14-Cre* strain.

### 4.2. Genotyping

Genotyping was performed using short-range end-point Polymerase Chain Reaction (PCR) assays with specific primers according to the EUCOMM/KOMP-CSD allele conversion guide. Genomic DNA was extracted from tail tips or punched ear tissues, followed by PCR analysis using the GoTaq Green master mix (Promega, Madison, WI, USA). The PCR products were separated on 2% agarose gels dissolved in Tris/Acetic Acid/EDTA (TAE) buffer and stained with ethidium bromide (Millipore-Sigma, St. Louis, MO, USA). Primer sequences are listed in Appendix A.

### 4.3. MC903-Induced AD-like Model

MC903 (calcipotriol hydrate) was purchased from Millipore-Sigma (Saint Louis, MO, USA). Both ears of 8–10-week-old mice were topically painted with 1 nmol of MC903 (in 20 μL ethanol; 10 μL on each side) on days 0, 1, 2, 3, 9, and 11. Ear thickness was measured with a digital caliper (Mitutoyo Corp., Tokyo, Japan).

### 4.4. X-Gal Staining

According to standard procedures, β-galactosidase activities encoded by the *Rora* promoter-driven LacZ gene in *Rora^tm1b/+^* mice were evaluated for the frozen ear sections. Frozen sections (8 μM) were fixed with 2% PFA/Phosphate Buffer Saline (PBS) containing 0.02% NP-40 at 4 ℃ for 30 min. After washing with the PBS buffer containing 0.02% NP-40 and 0.01% deoxycholate, the samples were incubated in X-gal reaction buffer (5 mM K_3_Fe(CN)_6_, 5 mM K_4_Fe(CN)_6_, 0.02% NP-40, 0.01% deoxycholate, 2 mM MgCl_2_, 5mM EGTA, and 1 mg/mL X-gal) at room temperature for at least 2 h. The sections were counterstained with Nuclear Fast Red. All reagents in this assay were purchased from Millipore-Sigma (St. Louis, MO, USA).

### 4.5. Histology and Immunofluorescence

Frozen sections (8 μM) of ear samples were fixed with 10% formalin for Hematoxylin and Eosin Y (H&E) staining and toluidine blue staining (Electron Microscopy Sciences, Hatfield, PA, USA). For immunostaining, frozen sections were fixed with 4% paraformaldehyde/PBS, permeabilized with 0.1% NP-40/PBS, and incubated with the appropriate primary antibody at 4 °C overnight. The slides were then incubated with Alexa 488- or Alexa 594-conjugated secondary antibodies (Invitrogen-Thermo Fisher Scientific, Waltham, MA, USA) and Hoechst (Invitrogen-Thermo Fisher Scientific) for DNA detection. H&E and fluorescence images were acquired using a Lionheart FX microscope (Biotek, Winooski, VT, USA). The number of dermal infiltrates (mast cells, CD11c+ positive cells, and CD4+ T cells) and the intensity of the epidermal TSLP signal were quantified using the ImageJ 1.53k software (National Institute of Health). A list of the primary antibodies is provided in Appendix A.

### 4.6. Quantitative Real-Time (RT)-PCR

According to the manufacturer’s instructions, total RNA from ear tissues was isolated using Trizol (Invitrogen-Thermo Fisher Scientific). Reverse transcription of isolated RNA was done using the High-Capacity cDNA RT Kit (Applied Biosystems-Thermo Fisher Scientific, Foster City, CA, USA). Amplification reactions were performed using PowerSYBR Green PCR Master Mix and a QuantStudio 3 Real-Time PCR system (Applied Biosystems-Thermo Fisher Scientific). Each sample was tested in duplicates. A comparative method for relative quantification (2^XΔΔ^ cycle threshold) was used to calculate the expression levels of the target genes, which were normalized to the level of 18S rRNA. The primer sequences are listed in Appendix A.

### 4.7. ELISA

Peripheral blood was collected to measure the serum IgE and TSLP levels at specific time points and analyzed using ELISA with commercial kits from eBioscience-Thermo Fisher Scientific according to the manufacturer’s instructions. For measuring TSLP protein levels from the ear skin, ear tissue was processed with a homogenizer (Fisherbrand Bead Mill 4, Fisher Scientific, Hanover Park, IL, USA) in protein lysis buffer (100 mM Tris, pH 7.4, 2 mM Na_3_VO_4_, 100 mM NaCl, 1% Triton X-100, 1 mM EDTA, 10% glycerol, 1 mM EGTA, 0.1% SDS, and protease inhibitors). For measuring IL13 secretion from ear-draining lymph nodes, single-cell suspension from the isolated lymph nodes was plated at a density of 5 × 10^5^ cells/well in 96-well plates. After 24 h of re-stimulation with 50 ng/mL TPA and 1 μM ionomycin, the media were collected to measure cytokine production using ELISA.

### 4.8. Transepidermal Water Loss Measurements

Transepidermal water loss (TEWL) was measured on ear skin before and after MC903 application using a noninvasive Tewameter TM Hex probe (Courage and Khazaka Electronic GmbH, Cologne, Germany) according to the manufacturer’s instructions.

### 4.9. Western Blot Analysis

The ear tissue was processed in protein lysis buffer using a tissue homogenizer, as described in the ELISA assay. Protein concentration was determined using the BCA Protein Assay Kit (Pierce-Thermo Fisher Scientific). Proteins were separated using SDS-polyacrylamide gel electrophoresis and transferred onto 0.45 μm polyvinylidene difluoride (PVDF) membranes (Thermo Fisher Scientific). After blocking with 5% non-fat milk, the membranes were incubated with specific antibodies (see Appendix A) overnight at 4 °C, followed by incubation with horseradish peroxidase (HRP)-conjugated secondary antibodies. The membranes were incubated with the enhanced chemiluminescence (ECL) western blotting substrate (Pierce Biotechnology-Thermo Fisher Scientific). Chemiluminescent images were acquired using an iBright CL1000 (Invitrogen—Thermo Fisher Scientific, Pittsburgh, PA, USA).

### 4.10. Statistical Analysis

All statistical evaluations used Prism 9.5 (GraphPad Software, La Jolla, CA, USA). Unpaired Student’s *t*-tests were used to compare the statistical difference between the two groups. The statistical differences between multiple groups were analyzed using a one-way ANOVA or two-way ANOVA, as indicated in the figure legends, and *p* values < 0.05 were considered significant.

## Figures and Tables

**Figure 1 ijms-24-10241-f001:**
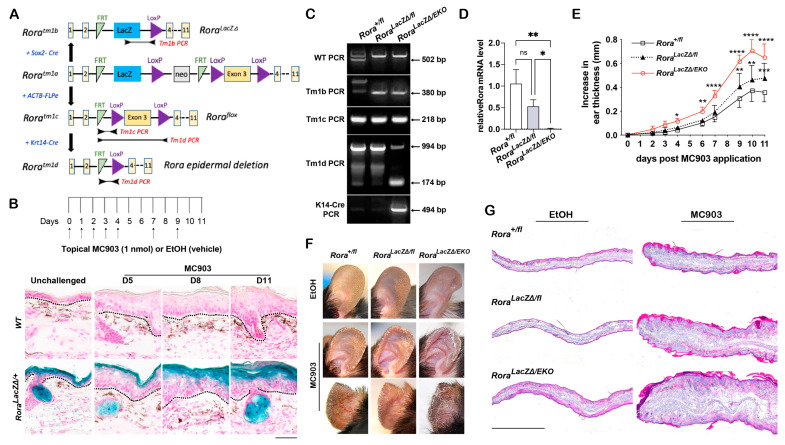
Epidermal *Rora* expression in response to MC903 and the effect of epidermal *Rora* deletion on MC903-induced ear swelling in *Rora^LacZ^^Δ/EKO^* mice. (**A**) Scheme of the targeted strategies for generating *Rora* mutant strains from the *Rora^tm1a^^/+^* strain. (**B**) Top: scheme of the MC903 treatment regimen. Arrows indicate the days when both ears were topically treated with 1 nmol of MC903 or 100% ethanol (EtOH). Bottom: X-gal staining (blue) of frozen ear skin sections from wildtype (WT) or the *Rora^LacZ^^Δ/+^* reporter mice collected at the indicated time points after topical MC903 application. DNA was counterstained with Nuclear Fast Red (pink). Black dotted lines mark the dermal–epidermal junctures; scale bar = 100 μm. (**C**) Validation of epidermal *Rora* depletion in *Rora ^LacZ^^Δ/EKO^* mice using short-range end-point PCR assays. The PCR products were separated using a 2% agarose gel and stained with ethidium bromide. (**D**) Total RNA was isolated from the adult ear skin of each genotype, followed by RT-PCR analysis of *Rora* mRNA normalized to *18S* rRNA. Values from MC903-treated ears were normalized to ethanol (EtOH)-treated control ears (set as 1) and presented as mean fold ± SD, *n* = 5/genotype. (**E**) Mouse ear thickness was measured at the indicated time points and plotted as the increase in ear thickness after subtracting the value of day 0; *n* = 8–10/genotype from 3 independent experiments. (**F**) Representative photos of mouse ears collected on day 11. (**G**) Representative images of the hematoxylin and eosin (H&E) staining on frozen sections of ear samples collected on day 11; scale bar = 1000 μm. *, *p* < 0.05, **, *p* < 0.01, ***, *p* < 0.001, ****, *p* < 0.0001, or ns (not significant), were determined through a one-way ANOVA (**D**) or a two-way ANOVA (**E**).

**Figure 2 ijms-24-10241-f002:**
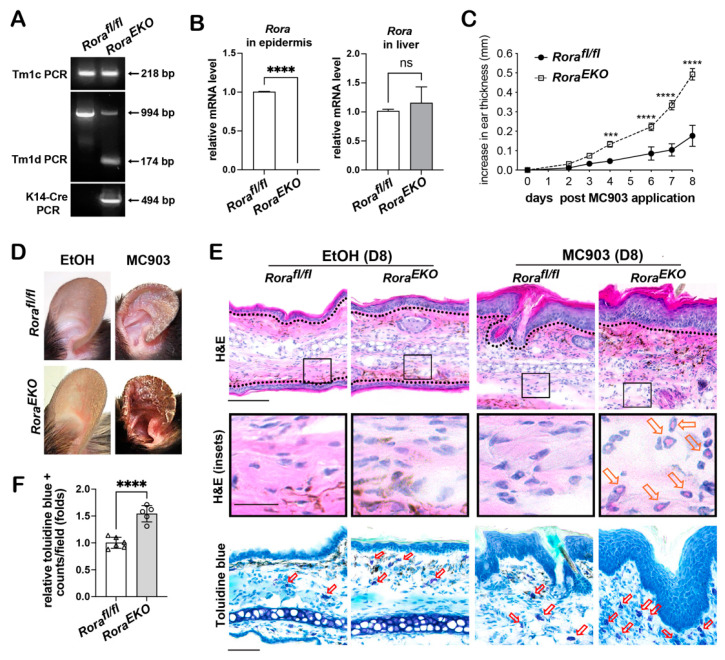
Epidermal *Rora* deletion exaggerates MC903-induced ear swelling in *Rora^EKO^* mice. (**A**) Validation of epidermal *Rora* depletion in *Rora^fl/fl;^K14-CRE (Rora^EKO^)* mice using short-range end-point PCR assays. (**B**) Total RNA was isolated from the back skin epidermis or the liver of postnatal day 4 (P4) mice, followed by RT-PCR analysis of *Rora* mRNA, as described in Figure 1D. (**C**) Mouse ears were topically treated with EtOH or MC903, as described in Figure 1B. Mouse ear thickness was measured and plotted, as described in Figure 1E; *n* = 8/genotype from 3 independent experiments. (**D**) Representative photos of mouse ears collected on day 11. (**E**) Top two panels show representative H&E images of frozen ear sections collected on day 8. The insets show eosinophils (arrows) in the dermis. The lower panel shows the representative images of toluidine blue (TB) staining of the frozen ear sections. Individual mast cells with intensive purple–blue signals are denoted by red arrows; scale bar = 100 μm (top and bottom panels); scale bar = 25 μm (insets). (**F**) The number of mast cells/field of vision under 20× magnification was quantified in MC903-treated ears (day 8). The average value was obtained for each ear section from 9–10 fields. Values from *Rora^EKO^* mice were normalized to the *Rora^fl/fl^* group and presented as mean fold ± SD, *n* = 5–6 from 3 mice/group. ***, *p* < 0.001, ****, *p* < 0.0001, or ns (not significant), were determined by an unpaired *t*-test.

**Figure 3 ijms-24-10241-f003:**
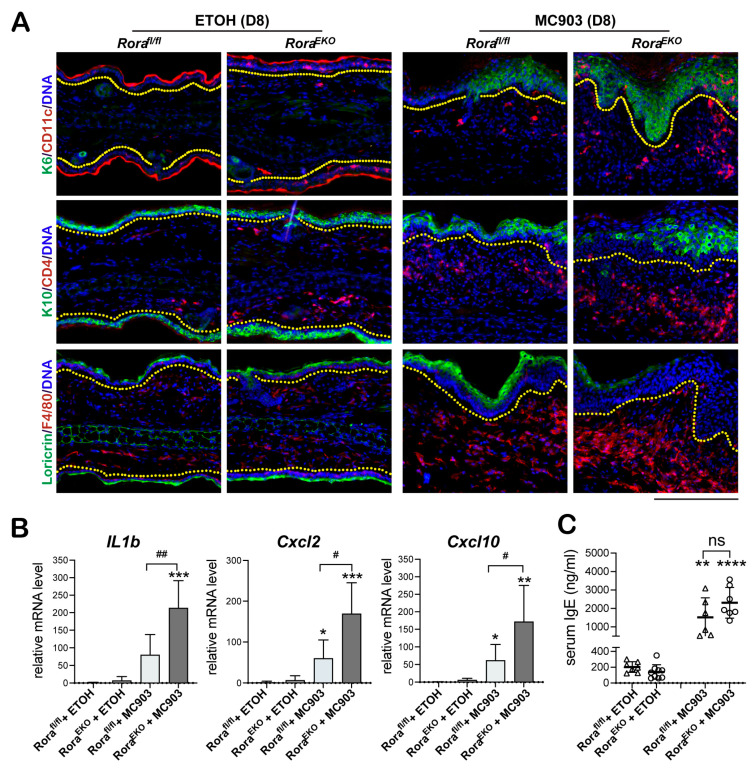
Epidermal *Rora* deletion aggravates MC903-induced AD-like symptoms in *Rora^EKO^* mice. Ear samples were collected on day 8 post-treatment. (**A**) Representative images of ear sections stained with antibodies against keratin 6 (K6, green) + CD11c (red), keratin 10 (K10, green) + CD4 (red), and Loricrin (green) + F4/80 (red). DNA was counterstained with Hoechst (blue). Dotted lines mark the dermal–epidermal junctures; scale bar = 200 μm. (**B**) mRNA expression of the indicated inflammatory factors was measured using quantitative RT-PCR and normalized to *18S* for each sample. Values were normalized to EtOH-treated *Rora^fl/fl^* ears and presented as mean fold ± SD, *n* ≥ 4/group. (**C**) Serum IgE levels of EtOH- or MC903-treated mice were measured using ELISA. *, *p* < 0.05, **, *p* < 0.01, ***, *p* < 0.001, or ****, *p* < 0.0001, was determined by one-way ANOVA; #, *p* < 0.05, ##, *p* < 0.01, or ns (not significant), indicate the difference between *Rora^fl/fl^* and *Rora^EKO^* mice within the MC903 treatment group.

**Figure 4 ijms-24-10241-f004:**
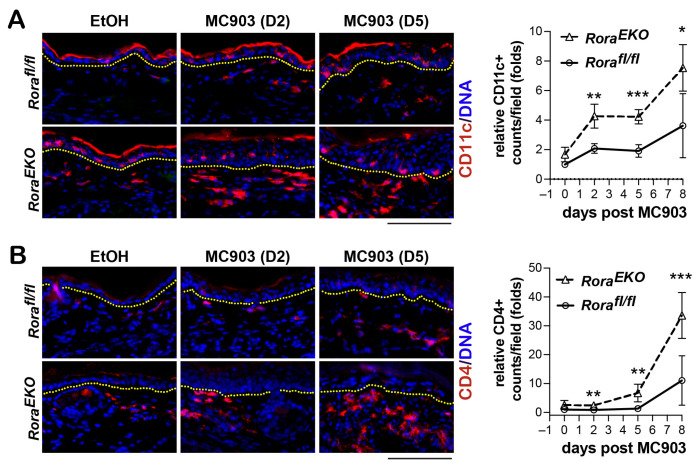
Epidermal loss of RORα accelerates MC903-induced dermal activation of dendritic cells and CD4+ T cells. Mouse ears were challenged with EtOH or MC903 and collected on day 2 or 5. (**A**,**B**) Representative immunostaining images with antibodies against CD11c (red) (**A**) or CD4 (red) (**B**); DNA was counterstained with Hoechst (blue); scale bar = 100 μm. Dotted lines mark the dermal–epidermal junctures. The number of CD11c or CD4-positive counts/field of vision under 20× magnification was quantified for 9–10 fields per ear section to obtain the average value. All values were normalized to an average value of the EtOH-treated *Rora^fl/fl^* group and presented as mean fold ± SD, *n* ≥ 3 for 3 mice/group. *, *p* < 0.05, **, *p* < 0.01, or ***, *p* < 0.001, were determined by two-way ANOVA.

**Figure 5 ijms-24-10241-f005:**
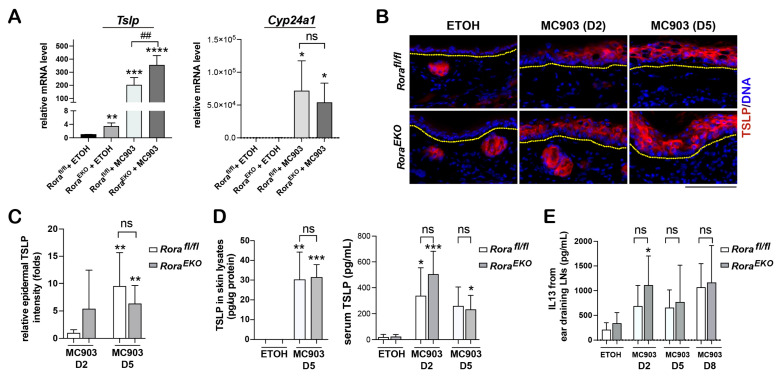
Epidermal *Rora* deletion on *Tslp* gene expression in the skin. (**A**) *Tslp* and *Cyp24a1* mRNA expression in ear samples collected on day 2 after EtOH- or MC903-treatment was measured using RT-PCR analysis. Values were normalized to EtOH-treated control ears and presented as mean fold ± SD, *n* = 4/group. (**B**) Representative images of frozen ear sections stained with an antibody against TSLP (red); DNA was counterstained with Hoechst (blue); scale bar = 100 μm. Dotted lines mark the dermal–epidermal junctures. (**C**) Epidermal TSLP Fluorescence intensity was quantified using ImageJ. The average value for each skin section was obtained from ≥5 fields of vision under 10× magnification, normalized to the EtOH-treated *Rora^fl/fl^* group, and presented as mean ± SD, *n* = 5–6 from 3 mice/group. (**D**) TSLP protein levels of the skin lysates and the serum were measured using ELISA. (**E**) IL13 secreted by the ear-draining lymph nodes was measured using ELISA. Single-cell suspension was prepared from the ear skin-draining lymph nodes and re-stimulated with PMA/ionomycin for 24 h. The supernatant was collected for IL13 analysis. Values are presented as mean ± SD, *n* ≥ 4/group. *, *p* < 0.05, **, *p* < 0.01, ***, *p* < 0.001, or ****, *p* < 0.0001, was determined by one-way ANOVA; ##, *p* < 0.01, or ns (not significant), indicate the difference between *Rora^fl/fl^* and *Rora^EKO^* mice within each of the MC903 treatment groups.

**Figure 6 ijms-24-10241-f006:**
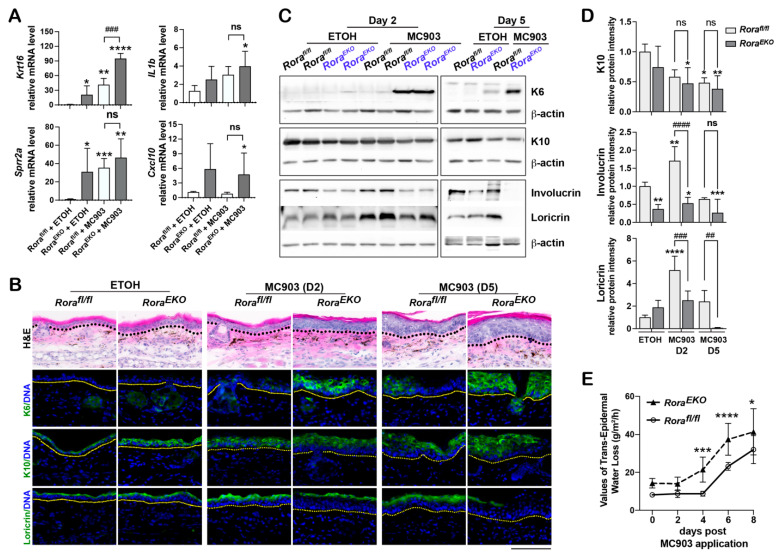
Epidermal loss of RORα deregulates keratinocyte differentiation and accelerates skin barrier disruption in response to MC903. (**A**) mRNA expression of indicated genes in the ear samples collected on day 2 after EtOH- or MC903-treatment; *n* = 4/group. (**B**) Representative images of H&E staining or immunostaining of frozen ear sections with antibodies against keratin 10 (K10, green), keratin 6 (K6, green), or loricrin (green). DNA was counterstained with Hoechst (blue); scale bar = 100 μm. Dotted lines mark the dermal–epidermal junctures. (**C**) Western blot analysis of indicated proteins in ear samples collected on day 2 after EtOH- or MC903-treatment. (**D**) The protein levels of indicated genes were quantified by densitometry scanning and normalized to b-tubulin. Values are presented as mean-fold over EtOH-treated *Rora^fl/fl^* control ± SD, *n* = 3–5/group. (**E**) Trans-epidermal water loss (TEWL) was measured on ear skin before and at different time points after MC903 application. *, *p* < 0.05, **, *p* < 0.01, ***, *p* < 0.001, or ****, *p* < 0.0001, was determined by one-way ANOVA (**A**) or two-way ANOVA (**D**,**E**); ##, *p* < 0.01, ###, *p* < 0.001, ####, *p* < 0.0001, or ns (not significant), indicate the difference between *Rora^fl/fl^* and *Rora^EKO^* mice within each of the MC903 treatment groups.

## Data Availability

Data sharing is not applicable as no datasets have been generated or analyzed in this study.

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
