# Peer review of "Epidermal Loss of RORα Enhances Skin Inflammation in a MC903-Induced Mouse Model of Atopic Dermatitis"

_ijms, 2023, doi:10.3390/ijms241210241_

Round 1
Author Response
Major Issues:
- In Figure 5, the authors conclude that, "enhanced activation of dermal dendritic cells in RoraEKO mice at the early stage post-MC903 treatment may be marginally attributed to altered TSLP expression.” However, other than increased mRNA levels of Tslp, the protein data for TSLP levels does not support a role of TSLP in this process. TSLP is only increased in the serum two days after MC903 treatment in RoraEKO mice and TSLP expression is actually decreased in the ear skin sections after five days of MC903 treatment in the RoraEKO mice. Since the TSLP protein and not mRNA activates dermal dendritic cells, this enhanced activation is likely TSLP-independent. Other mechanisms of MC903-initiated inflammation should be considered and discussed.
Response 1: We appreciate the reviewer's suggestion and have made changes accordingly (please see line 233 on Page 7). We have also added a paragraph in the Discussion section to confer the potential mechanisms for enhanced inflammation in RoraEKO mice (please see lines 311-320 on Page 9).
- In Figure 6, the authors conclude that epidermal RORα depletion causes a decrease in epidermal differentiation. They reach this conclusion due to the protein expression of Keratin 10, Involucrin, and Loricrin. However, in Figure 6C, the loading control, β-actin, is not equal in the samples, and Keratin 10 is overexposed. The samples should either be run so that the protein loading is equal throughout the samples or quantification should be down to determine if differentiation markers are significantly decreased in epidermal RORα knockout skin with MC903 treatment.
Response 2: This is a very important point. We have re-run the samples for better keratin 10 detection and added the immunoblotting results of day 5 samples in the new Figure 2C. Protein levels were quantified by densitometry scanning using the ImageJ software (new Figure 2D). As shown in the revised Figure 6C-D, epidermal Rora deletion had a marginal effect on keratin 10 expression but significantly reduced loricrin and involucrin protein levels upon MC903 treatment. Please see lines 270-280 on Pages 8-9 for details of these results.
Minor Issues:
- In Figure 1B, it is difficult to distinguish the epidermal-dermal junctures. Addition of dotted lines, like that in Figure 3A, would be beneficial.
Response 1: We have added dotted lines to mark the epidermal-dermal junctures in Figure 1B.
- For Figure 2B, insets for the dermal eosinophils should be shown for all the conditions for comparison. It is difficult to see any differences in the representative panels that are currently shown.
Response 2: Great Suggestion! We have removed the original top panel of HE images taken at low magnification and added a new panel (middle) of insets for all 4 conditions in the new Figure 2E.
Reviewer 2 Report
The manuscript "Epidermal loss of RORa enhances skin inflammation in 2 MC903-induced mouse model of atopic dermatitis" is a very valuable manuscript in which the authors present their original research results. Thus, in this study, they examined the roles of epidermal RORa in regulating AD pathogenesis, where they generated mouse strains with epidermis-specific Rora ablation. According to their results, Rora deficiency greatly amplified MC903-elicited AD-like symptoms influencing on various factors - by intensifying skin scaliness, increasing epidermal hyperproliferation and barrier impairment, and elevating dermal immune infiltrates (proinflammatory cytokines and chemokines). In addition, despite the normal appearance at the steady state, Rora deficient skin showed microscopic abnormalities (including mild epidermal hyperplasia, increased TEWL, and elevated mRNA expression of Krt16, Sprr2a, and Tslp genes), indicating the subclinical impaired epidermal barrier functions. So, their results substantiate the importance of epidermal RORa in suppressing AD development partially by maintaining normal keratinocyte differentiation and skin barrier function.
The authors presented their results very comprehensively and used many visual presentations and tables - which contribute to quality presentation.
There are only some observations:
In Discussion: Limitations and strengths of the study should be mentioned.
Also, if possible, the authors could mention additional findings obtained by other studies on the same molecules.
Can the authors mention more data on potentially application of their findings for further therapy options and the practical work?
Also, cited references should be written in one uniform style - e.g. the titles of references are sometimes by upper, sometimes by lower letters.
Author Response
- In Discussion: Limitations and strengths of the study should be mentioned.
Response 1: We have added the limitations and strengths of the study in the last paragraph of the Discussion section (please see lines 364-368 on Page 10).
- Also, if possible, the authors could mention additional findings obtained by other studies on the same molecules.
Response 2: We appreciate this valuable suggestion and have included more background knowledge on RORa from other studies. This information is added in the Introduction section (please see lines 51-76 on Page 2).
- Can the authors mention more data on the potential application of their findings for further therapy options and the practical work?
Response 3: One potential application of our findings is that RORa agonists could be beneficial for restoring barrier functions and alleviating inflammation in treating AD and other inflammatory skin diseases. We have added new material to the Discussion section (please see lines 362-364 on Page 10).
- Also, cited references should be written in one uniform style - e.g., the titles of references are sometimes by upper, sometimes by lower letters.
Response 4: This problem has been fixed.
Reviewer 3 Report
In the paper entitled “Epidermal loss of RORa enhances skin inflammation in MC903-induced mouse model of atopic dermatitis.”, the authors identified epidermal retinoid-related orphan nuclear receptor (RORα) as a novel transcription factor critical for maintaining normal epidermal barrier functions and suppressing atopic dermatitis development.
The paper is well written and informative, however, the experiments were conducted under non physiological conditions using the epidermal specific Rora knockout mice.
The authors will be asked to answer the most critical question “What causes downregulation of RORα ?”.
Author Response
Reviewer 3:
The paper is well written and informative, however, the experiments were conducted under nonphysiological conditions using the epidermal specific Rora knockout mice.
The authors will be asked to answer the most critical question, "What causes downregulation of RORα ?"
Response: We appreciate this important question. It is unclear what causes the downregulation of RORa. But it has been found that Th2 cytokines can modulate the activity of other transcription factors, such as p63 and Notch, thereby downregulating the expression of keratinocyte differentiation markers. We believe that RORa expression is also under the negative control of proinflammatory cytokines, although the detailed mechanism remains unelucidated. We have included this discussion in lines 329-336 on Page 10, and addressing this issue will be part of our future studies.